# Assessment of Semen Respiratory Activity of Domesticated Species before and after Cryopreservation: Boars, Bulls, Stallions, Reindeers and Roosters

**DOI:** 10.3390/vetsci9100513

**Published:** 2022-09-21

**Authors:** Elena Nikitkina, Ismail Shapiev, Artem Musidray, Anna Krutikova, Kirill Plemyashov, Sofia Bogdanova, Victoria Leibova, Gennadiy Shiryaev, Julia Turlova

**Affiliations:** Russian Research Institute of Farm Animal Genetics and Breeding—Branch of the L.K. Ernst Federal Research Center for Animal Husbandry, 55A, Moskovskoye sh., Tyarlevo, 196625 Pushkin, Russia

**Keywords:** semen, oxidative phosphorylation, 2.4-dinitrophenol, fertilizing ability, cryopreservation

## Abstract

**Simple Summary:**

Artificial insemination is actively used in animal husbandry. It is important to know the quality of the sperm for artificial insemination. One of the indicators of sperm quality can be an assessment of energy metabolism, since energy is needed for sperm to move and fertilize the egg. We studied the respiration rate in spermatozoa of different animal species: bulls, stallions, boars, reindeer and roosters. To determine the production of energy (ATP), the substance 2.4-dinetrophenol (2.4-DNP) was used, which stopped the production of ATP. Semen was assessed before and after freezing. The evaluation showed the same response to the addition of 2.4-DNP to the semen of different species, as well as a sufficient relationship between the reaction of semen respiration to the addition of 2.4-DNP and the fertilizing ability of sperm. At the same time, no relationship was found between the respiratory rate and fertility. The 2.4-DNP test can be a suitable additional measure of sperm quality.

**Abstract:**

To assess sperm quality, it is important to evaluate energy metabolism. The test substance 2.4-dinitrophenol (2.4-DNP) is an agent for destroying oxidative phosphorylation. 2.4-DNP shuts off the production of adenosine triphosphate (ATP) from oxidation and then, the respiration rate increases. If the respiratory chain is damaged, there is little or no response to adding 2.4-DNP. The aim of this study was to analyze the respiratory activity and oxidative phosphorylation in semen before and after freezing and compare the obtained data with the fertilizing ability of sperm. There was a reduction in sperm respiration rates in all species after thawing. The respiration of spermatozoa of boars, bulls, stallions, reindeers and chicken showed responses to 2.4-dinitrophenol. The only difference is in the strength of the response to the test substance. After freezing and thawing, respiratory stimulation by 2.4-DNP decreased. The results of our study show that respiration rate is not correlated with pregnancy rates and egg fertility. However, there was a high correlation between the stimulation of respiration by 2.4-dinitrophenol and pregnancy rates. The test for an increase in respiration rate after adding 2.4-dinitrophenol could be a suitable test of the fertilizing ability of sperm.

## 1. Introduction

The central place in the vital activity of cells is occupied by biochemical processes, namely oxidative phosphorylation and glycolysis, associated with the generation of energy (adenosine triphosphate-ATP). Oxidative phosphorylation predominates over glycolysis and is the main source of ATP synthesis. Therefore, a violation of the process of oxidative phosphorylation, during freezing and thawing of spermatozoa, can become the main reason for a decrease in their fertilizing ability [1,2,3].

Spermatozoa require energy to perform tasks such as motility and interaction with oocytes [3,4,5,6]. Many studies have been devoted to spermatozoa mitochondria [3,7,8,9,10,11]. Some species, such as sheep and cattle, are reported to use glycolysis more, while horses and pigs mainly use oxidative phosphorylation [12,13,14]. Little is known about cock semen, and nothing is known about reindeer semen.

Numerous studies have established that the ratio of respiration and phosphorylation is a sensitive indicator of the functional state of the cell and the whole organism [2,13,15,16]. The intensity of oxidative phosphorylation is an objective criterion for assessing the state of spermatozoa under various conditions of preserving the sperm of farm animals.

To reduce damage and increase the lifespan of spermatozoa during cryopreservation, special media are needed to provide energy [2,17]. An assessment of the state of oxidative phosphorylation is important in assessing the quality of animal and human semen.

Polarography is a method for determining the concentration of substances in a solution by measuring the current–voltage curve of polarized electrodes during electrolysis [18]. We used the Clark electrode. The Clark electrode uses a combination of a polarizable electrode usually consisting of platinum in combination with an Ag/AgCl electrode. Each O_2_ molecule at the electrode generates four electrons in the electrode that can be measured as a current, directly proportional to the amount of oxygen in the solution. The polarographic method for determining the oxygen concentration is based on the reaction of oxygen reduction at the cathode at a position of a potential difference of 0.6–0.8 V. This occurs when the current is a measure of oxygen concentration. Respiration is recorded as a decrease in diffuse current in the process of a decrease in oxygen absorbed by the object. The polarographic method we developed makes it possible to assess the functional state of the energy system, particularly the conjugation of respiration with the synthesis of ATP. There is a clear correlation between the total ATP content and the integrity of sperm membranes [19,20]. This indicator reflects one of the mechanisms of regulation of energy metabolism.

The aim of this study was to evaluate the respiratory activity and oxidative phosphorylation in semen before and after freezing and to assess the relationship between the rate of respiration, the 2.4-dinitrophenol test and the fertilizing ability of sperm.

## 2. Materials and Methods

### 2.1. Ethics Statement

The principles of laboratory animal care were followed, and all procedures were conducted according to the ethical guidelines of the L.K. Ernst Federal Science Center for Animal Husbandry. The protocol was approved by the Commission on the Ethics of Animal Experiments of the L.K. Ernst Federal Science Center for Animal Husbandry (protocol number: 2020/2) and the Law of the Russia Federation on Veterinary Medicine No. 4979-1 (14 May 1993).

### 2.2. Animals

Large white pigs, Hanoverian and Arabian horses, Holstein bulls, Nenets reindeers and Leghorn roosters were used in this study. All studied animal species were kept in conditions in accordance with the veterinary standards of the Russian Federation.

### 2.3. Chemicals and Extenders Preparation

The chemicals used for the preparation of freezing extenders, 2.4-dinitrophenol and lactose, were ordered from Sigma-Aldrich (Sigma, Saint Louis, MO, USA). Egg yolk was collected from chickens of the bioresource collection “Genetic Collection of Rare and Endangered Chicken Breeds” (RRIFAGB, Pushkin, St. Petersburg, Russia). The centrifugation solution for equine semen consisted of 204 mM lactose, 25 mM glucose, 3 mM ethylenediaminetetraacetic acid dinatrium salt (Trilon B), 0.4 mM magnesium sulfate, 21 mM sodium chloride and 14 mM potassium citrate. The components of the Leningrad Cryoprotective Medium (LCM) for cock semen freezing (Tselutin 2013) per 100 mL of distilled water were as follows: 114 mM monosodium glutamate, 44 mM fructose, 51 mM potassium acetate, 8.3 µM polyvinylpyrrolidone and 3.27 µM protamine sulfate.

### 2.4. Semen Collection and Preparation

Bull, horse and boar semen samples were collected with an artificial vagina. Rooster semen was collected by the abdominal massage technique. Reindeer semen was collected by electroejaculation. Sperm concentration, total motility (TM) and progressive motility (PM) were evaluated by computer-assisted sperm analysis (CASA) in a Makler chamber at 37 °C. The Argus CASA system (ArgusSoft LLC., St. Petersburg, Russia) and a Motic BA 410 microscope (Motic, Hong Kong, China) were used. The semen was frozen according to protocol for each animal.

### 2.5. Semen Freezing

#### 2.5.1. Boar Semen Freezing

Fresh boar semen was diluted 1:1 in GHUJKUM medium (inventor certificate no. 540634, Russia) and cooled to 22 °C for 2 h. It was then centrifuged for 15 min at 700× *g*, the supernatant was removed and the pellet was resuspended to a final concentration of 1.5 billion spermatozoa/ml and cooled to 4–5 °C at 0.2 °C/min. Prior to freezing the pellets, glycerol was added at a final concentration of 2%. Then, samples were frozen in liquid nitrogen vapor for 10 min.

#### 2.5.2. Equine Semen Freezing

The diluted 1:1 equine samples were centrifuged for 8 min at 600× *g*, the supernatant was eliminated and samples was resuspended in Steridyl^®^ medium (Minitüb GmbH, Tiefenbach, Germany). The final concentration was 200 × 10^6^ cells/mL. Semen was loaded into 0.5 mL straws and equilibrated at +5 °C for 120 min. The straws were frozen in liquid nitrogen vapor at −110 °C for 12 min, and stored in a liquid nitrogen tank.

#### 2.5.3. Bovine Semen Freezing

Fresh bovine semen was diluted in Steridyl^®^ medium (Minitüb GmbH, Tiefenbach, Germany) to a final concentration of 150 × 10^6^/mL, loaded into 0.25 mL straws, equilibrated at +5 °C for 4 h and then frozen in liquid nitrogen vapor for 10 minutes before being submerged.

#### 2.5.4. Reindeer Semen Freezing

Fresh reindeer semen was extended in Steridyl to a final concentration of 100 × 10^6^/mL. Reindeer semen was loaded into 0.25 mL straws and equilibrated for 2 h at 5 °C. Straws were frozen in liquid nitrogen vapor for 10 min before being submerged.

#### 2.5.5. Roosters Semen Freezing

Diluted rooster semen samples were equilibrated from 18 °C to 5 °C for 40 min. After cooling, dimethylacetamide (DMA, Sigma Aldrich, St. Louis, MO, USA) was added to each sample at a final concentration of 6%. After adding DMA, the samples were incubated at 5 °C for 1 min. Freezing was carried out in pellets by directly dripping the semen into liquid nitrogen.

### 2.6. Semen Evaluation after Thawing

After at least 24 h, stallion, bull, boar and reindeer semen samples were thawed at 37 °C for 30 s, and the contents of the straw were emptied into a 1.5 mL microcentrifuge tube. Boar semen was thawed at 37 °C for 30 s, and then the sample was emptied into a 1.5 mL microcentrifuge tube. Rooster semen was thawed on a heated metal plate at 60 °C and then emptied into a 1.5 mL microcentrifuge tube. Semen was evaluated for progressive motility as previously described.

We evaluated the respiration rate (CR) using the Expert-001MTX ion meter and a Clarke electrode (Research and Production Company “Econix-Expert”, Moscow, Russia) [21]. Next, 100 μL of semen was added to the chamber with 1 mL of 11% lactose, and the rate of decrease in oxygen concentration was measured. Then, 10 μL 2.4-DNP was added. The ratio of the respiration rate with 2.4-DNP to the respiration rate before adding 2.4-DNP was determined. This was a respiration reaction to the addition of 2.4-DNP and was equal to 1 or more.

### 2.7. Artificial Insemination (AI) and Pregnancy Control

Fifty sows were inseminated with frozen semen. Pregnancy was determined by ultrasound examination on the 40th day after ovulation. Fifty cows were inseminated with frozen sperm with different stimulation of respiration by 2.4-dinitrophenol. Pregnancy was determined by ultrasound examination on the 30th day after ovulation. To test the rooster sperm fertility, hens were inseminated intravaginally according to the following scheme: two days in a row with a single dose of insemination of 0.04–0.07 mL of frozen/thawed semen (insemination dose was at least 70–80 million progressively moving spermatozoa), and then every three days. The total number of insemination days was five. Collecting eggs for incubation began a day the after the first insemination and was performed daily for nine days. Eggs were incubated for six days to assess the fertility of frozen/thawed semen (n = 300 eggs). Egg fertility rates were determined by blastoderm development after artificial insemination.

### 2.8. Statistical Analysis

For statistical analysis, the computer software IBM-SPSS Statistics 19 (IBM, Armonk, NY, USA) was used. All data were normally distributed (Kolmogorov–Smirnov test). Data were analyzed by ANOVA. The data were expressed as means ± standard error of the mean. All pairwise multiple comparisons between means were conducted by *t*-tests. Differences were considered statistically significant at *p* < 0.05.

## 3. Results

The semen used in this study was in the acceptable range for sperm motility and morphology after dilution and preparation for freezing.

The assessment of respiratory activity before and after freezing is shown in Table 1.

There was a reduction in sperm respiration rates in all species after thawing. The respiration of spermatozoa of boars, bulls, stallions, reindeer and chickens showed the same reaction to 2.4-dinitrophenol. The only difference was in the strength of the response to the test substance. After freezing and thawing, respiratory stimulation by 2.4-dinitrophenol decreased.

Evaluation of the fertility of boar sperm showed a correlation with respiratory stimulation by 2.4-dinitrophenol (Table 2). One group of sows was inseminated with sperm with 1.4 ± 0.01 respiration stimulation by 2.4-dinitrophenol, and fertility was 35 ± 4.6%; a second group was inseminated with sperm with 1.8 ± 0.05 respiration stimulation by 2.4-dinitrophenol, and fertility was 53.8 ± 8.0%; and a third group was inseminated with sperm with 2.1 ± 0.05 respiration stimulation by 2.4-dinitrophenol, and fertility was 66.7 ± 6.0%. The correlation coefficient was high (r = 0.70, *p* < 0.01).

Pregnancy rates of bovine and cock semen with different stimulation rates by 2.4-dinitrophenol are presented in Table 3 and Table 4, respectively.

There was no correlation between pregnancy rate and progressive motility in bovine semen, but there was a correlation between pregnancy rate and stimulation of respiration by 2.4-dinitrophenol (r = 0.62, *p* < 0.05).

The same results were found with rooster semen (Table 4).

Respiration rate has no correlation with egg fertility. Thus, the sperm of males No. 7868 and No. 7872 with a similar fertilizing ability differs significantly in the respiration rate. Likewise, the semen of rooster No. 7864 and No. 7870 with similar respiration rates differs in fertilizing ability. Correlation analysis of the relationship between egg fertility rate during insemination with frozen–thawed semen and the stimulation of respiration by 2.4-dinitrophenol showed that this relationship is strong (r = 0.76, *p* < 0.01).

## 4. Discussion

The intensity of energy metabolism is an important criterion to assess sperm quality [16,21,22]. Oxidative phosphorylation is a metabolic pathway in which cells oxidize nutrients and produce adenosine triphosphate (ATP). Glycolysis predominates in ram semen. Stallion and boar spermatozoa mainly depend on oxidative phosphorylation for the production of ATP [11,13,21,23]. Guinea pig and boar spermatozoa are virtually unable to maintain motility through anaerobic glycolysis [3,24,25]. Bovine and rhesus monkey spermatozoa can meet their energy needs through both glycolysis and respiration [25]. It has been established that the rates of bull spermatozoa are comparable under aerobic and anaerobic conditions in a medium containing glucose [26]. Mitochondria are sensitive to the damaging effects of low temperatures [9,27].

Oxidative phosphorylation is associated with the transfer of electrons (protons) along the respiratory chain. Therefore, dysregulation of the electron (proton) transport system in the respiratory chain of mitochondria is one of the causes of cell death under the damaging effect of low temperatures. The main consumer of oxygen, the mitochondrial respiratory chain, is a set of sequential redox reactions that transfer hydrogen and electrons from the substrate to oxygen [2,4,14,15]. The functional state of the mitochondrial respiratory chain in spermatozoa was assessed by the respiration response to the addition of 2.4-dinitrophenol, a respiration uncoupler. 2.4-dinitrophenol is an organic compound with the formula C_6_H_4_N_2_O_5_. 2.4-dinitrophenol acts as a protonophore, allowing protons to leak across the inner mitochondrial membrane and thus bypass ATP synthase. This makes ATP energy production less efficient. The phosphorylation of adenosine diphosphate becomes disconnected from oxidation [20]. Oxidative phosphorylation was assessed by the reaction of the cellular respiration rate (CR) upon adding 2.4-dinitrophenol (2.4-DNP) to the semen sample [21]. The addition of 2.4-DNP disrupts the proton gradient by carrying protons across a membrane, and uncouples proton pumping from ATP synthesis because it carries them across the inner mitochondrial membrane. As a result, the respiration rate increases. If the respiratory chain is damaged, there is little or no response to 2.4-dinitrophenol.

The respiration rate response to the addition of 2.4-DNP to semen samples equal to 1 indicates that the respiratory chain is damaged and that there is no ATP synthesis. A value of 1 indicates that the respiratory rate did not increase after the addition of 2.4-dinitrophenol.

Our study shows that the respiration rate decreased after cryopreservation of semen from all animals and roosters. The respiratory response of an increased respiration rate to the addition of 2.4-dinitrophenol to semen samples was also similar in different animal species. The greatest response was in boars. It was confirmed that oxidative phosphorylation predominates in boars.

The lowest respiration response to the addition of 2.4-dinitrophenol was in reindeer semen. Since little is known about reindeer semen, we assume that in the process of ATP formation, glycolysis predominates [19]. Notwithstanding the low values of the increase in the respiration rate with the addition of 2.4 DNP, a correlation was found between this indicator in fresh semen and progressive activity after thawing (r = 0.65, *p* < 0.01). Therefore, the assessment of oxidative phosphorylation can serve as a biomarker for the quality of reindeer sperm and its cryopreservation ability.

The respiratory rate response to the addition of 2.4 DNP decreased after freezing in semen in all studied animals and roosters, which is an indicator of an increase in free oxidation without oxidative phosphorylation. Damage to the electron transport chain can lead to the formation of reactive oxygen species (ROS) [28,29]. ROS can cause lipid peroxidation, which damages the integrity and motility of spermatozoa [1,30,31]. ROS are involved in spermatozoa capacitation and acrosomal reaction, the fusion of spermatozoa and oocytes at low concentrations [32,33]. High levels of ROS have been associated with low sperm motility and infertility [11,34]. The production of ROS in semen has been classically considered a harmful by-product of mitochondrial metabolism, leading to cellular damage and oxidative stress [11,35]. Oxidative stress can lead to damage to sperm functions such as loss of motility, oxidative DNA damage and caspase activation, which can lead to apoptosis [11].

The difficulty in predicting the functional usefulness of fresh and frozen sperm lies in the dependence of the fertilizing ability on many factors. An accurate assessment can be achieved on the basis of a comprehensive study based on the sum of morphological, functional and biochemical parameters characterizing the state of the nucleus, acrosome, locomotor apparatus and the structural and functional state of the sperm membrane. Determining a large number of indicators for accurate prediction of the fertilizing ability of sperm is a laborious process. It is necessary to choose a minimum number of indicators or develop a universal test for an accurate assessment of sperm and predicting its fertility. The main indicator is sperm motility. However, as our studies have shown, the semen of bulls did not differ in mobility, but showed a different fertilizing ability.

The results of our study show that respiration rate is not correlated with pregnancy rates and egg fertility. There was high individual variability in the rate of respiration. There was no relationship between the rate of respiration and motility of sperm in all studied animals and roosters. This shows that respiratory rate is not a predictor of good fertility or cryopreservation ability.

However, there was a high correlation between the stimulation of respiration by 2.4-dinitrophenol and pregnancy rates. This is most likely due to the fact that with low stimulation of respiration with 2.4-dinitrophenol, the formation of ATP is impaired and the sperm does not have enough energy to reach the site of fertilization, capacitation and fertilize itself. Spermatozoa with impaired movement or depleted fuel resources lose their ability to move forward and cannot fertilize the egg [6]. Sperm ATP requirements change during maturation, capacitation and hyperactivation. At this time, spermatozoa require an increasing supply of ATP. Stimulation of respiration by 2.4-dinitrophenol shows whether ATP production was efficient before adding 2.4-dinitrophenol.

Our studies have shown that for effective fertilizing ability, the response of increased respiration rate to the addition of 2.4-dinitrophenol to semen samples should be above 1.6 in bovine semen, above 2.0 in boar semen and above 1.8 in cock semen. Further research is required to determine the optimal values of respiration rate stimulation by 2.4-dinitrophenol in semen of horses and reindeer for fertilizing ability.

## 5. Conclusions

Our studies have shown high individual variability in the respiration rate of sperm in all studied species. No correlation was found between respiratory rate and sperm fertility. Respiratory rate and respiratory rate response to the addition of 2.4-dinitrophenol to a semen sample decreased after sperm cryopreservation in all studied species. A correlation was found between the respiratory rate response to the addition of 2.4-dinitrophenol to a semen sample and fertility in bulls, boars and roosters.

The test for an increase in respiration rate after adding 2.4-dinitrophenol could be a suitable test of the ability of sperm to fertilize.

## Figures and Tables

**Table 1 vetsci-09-00513-t001:** Respiratory activity in sperm of boar, stallion, bull, reindeer and rooster semen before and after freezing.

Animal	Semen	n	Progressive Motility, %	Respiration Rate nAO_2_/min	Stimulation of Respiration by 2.4-dinitrophenol
Boar	Fresh	7	90.4 ± 3.67 ^a^	178 ± 18.6 (49–220)	3.30 ± 0.05 (1–4.5)
	Frozen	7	35.2 ± 2.86 ^b^	98 ± 11.4 (23–177)	2.45 ± 0.15 (1–3.8)
Stallion	Fresh	10	79.7 ± 5.45 ^a^	281 ± 68.0 (56–324)	1.80 ± 0.14 (1–3.2)
	Frozen	10	43.3 ± 6.73 ^b^	136 ± 115.0 (65–235)	1.50 ± 0.11 (1–2.3)
Bull	Fresh	10	81.1 ± 3.23 ^a^	132 ± 7.0 (28–210)	2.20 ± 0.35 (1–3.6)
	Frozen	10	50.2 ± 2.13 ^b^	87 ± 11.3 (23–156)	1.59 ± 0.07 (1–2.5)
Reindeer	Fresh	10	79.1 ± 5.45 ^a^	–	1.70 ± 0.08 (1–2.4)
	Frozen	10	30.4 ± 8.46 ^b^	–	1.30 ± 0.09 (1–1.9)
Rooster	Fresh	10	91.0 ± 2.19 ^a^	225 ± 21.2 (67–354)	1.87 ± 0.04 (1–5.7)
	Frozen	10	45.2 ± 2.36 ^b^	121 ± 12.6 (46–212)	1.66 ± 0.11 (1–4.6)

^ab^*p* < 0.001.

**Table 2 vetsci-09-00513-t002:** Pregnancy rates of boar semen with different stimulation rates by 2.4-dinitrophenol.

Pig Groups	Progressive Motility, %	Stimulation of Respiration by 2.4-dinitrophenol	Pregnancy Rates, %
Pregnancy rate < 50%	42.7 ± 0.53	1.4 ± 0.01	35 ± 4.6 ^a^
Pregnancy rate 50–60%	44.0 ± 0.67	1.8 ± 0.05	53.8 ± 8.0 ^b^
Pregnancy rate > 60%	43.7 ± 0.86	2.1 ± 0.05	66.7 ± 6.0 ^c^

^abc^*p* < 0.05.

**Table 3 vetsci-09-00513-t003:** Pregnancy rates of bovine semen with different stimulation rates by 2.4-dinitrophenol.

Cows Groups	Progressive Motility, %	Stimulation of Respiration by 2.4-dinitrophenol	Pregnancy Rates, %
Pregnancy rate<50%	50.5 ± 0.67	1.34 ± 0.077 ^d^	39.1 ± 3.6 ^a^
Pregnancy rate 50–60%	51.0 ± 0.46	1.41 ± 0.029 ^e^	55.3 ± 2.0 ^b^
Pregnancy rate>60%	51.1 ± 0.73	1.59 ± 0.025 ^f^	70.5 ± 1.6 ^c^

^abc^*p* < 0.05, ^df^
*p* < 0.05, ^ef^
*p* < 0.05.

**Table 4 vetsci-09-00513-t004:** Respiration activity in rooster semen and egg fertility rates.

Rooster, Individual No	Respiration Rate, nAO_2_/min	Stimulation of Respiration by 2.4-dinitrophenol	Egg Fertility Rates,%
7861	71.0	1.85	67
7863	59.0	1.62	44
7864	50.0	1.69	62
7868	86.0	1.72	72
7870	50.5	1.79	75
7871	42.0	1.88	82
7872	65.0	1.77	74
7873	79.0	1.87	89
7875	70.0	1.70	67
7899	34.0	1.49	53

## Data Availability

Data will be made accessible from corresponding authors upon reasonable request.

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
