# Peer review of "Assessment of Semen Respiratory Activity of Domesticated Species before and after Cryopreservation: Boars, Bulls, Stallions, Reindeers and Roosters"

_vetsci, 2022, doi:10.3390/vetsci9100513_

Round 1

Reviewer 1 Report (Previous Reviewer 2)

The authors have addressed most reviewer concerns, and the manuscript is suitable for publication in Veterinary Sciences. There are some unneccesary redundancies in the text (e.g., at line 136 uncoupling of proton gradient repeated in one sentence), and a few phrasing issues which can be taken care of with some English editing. This manuscript is suitable for publication now.

Author Response

Thank you for review

Reviewer 2 Report (New Reviewer)

The proposed MS entitled “Assessment of Semen Respiratory Activity of Domesticated Species Before and After Cryopreservation: boars, bulls, stallions, reindeers and roosters” examined the measurement of the respiratory activity in fresh and cryopreserved spermatozoa of different domesticated species as a tool to determine the sperm capability to fertilize. The effect of the compound 2,4-dinitrophenol (2,4-DNP) on the stimulation of respiration on the pregnancy outcome was also tested with frozen-thawed semen from pig, bull and rooster.

First, this MS is poorly written in regard to both the English quality, scientific presentation and formatting. For instance, the information conveyed in their introduction lacks organization and fluidity. The topics covered should go from general to specific, and conclude with a hypothesis followed by a brief summary of the methodology used to test the hypothesis; the latter was never mentioned in the MS. Not to mention that in many instances, the authors either misspelled words or did not use the correct terminology. Nowadays, there are plenty of grammar and spelling tools and translation services to avoid such mistakes. As a reviewer, I should only have to focus on the scientific content of the MS, not on the English. Moreover, the authors were inconsistent in terms of using abbreviations, sometimes using minutes or min (e.g., lines 102 and 106), hours or h (e.g., lines 116 and 129), and writing numbers with symbols with and without spacing (e.g., lines 110 and 115: 200 x 106/ml vs. 150 x106/ml). The same observations were made in the Results section. I could reject this MS solely based on its mediocre quality in terms of writing and formatting.

Second, I have several concerns about the scientific validity of the current study. I have detailed the most relevant issues below:

1)      This MS does not have a testable hypothesis as the authors only mentioned the aim for conducting their study. The complexity of their single aim is confusing and would require a multi-factorial experimental design to analyze and validate their results. For instance, t-tests and correlations are not adequate to analyze most of the data presented in this study.

2)      The Results were ineffectively presented and the statistical analyses were not always shown and/or adequate for the type of experimental design of the study. 

i.         In a scientific paper, the Tables should ONLY be used if the data cannot be presented graphically (visual portion of the Results section). The main issue is that some of their data are not analyzed properly, leading to the misinterpretation of their data. As a consequence, their conclusions are not fully supported by the results provided. For example, the authors talked about correlations in the Results section and no correlation tests have been described in the Materials and Methods section, with no graphs showing these results either. This study examined the effects of several factors (cryopreservation, respiration rate, stimulation of respiration by 2,4-DNP) on a response variable (sperm fertility). By not following a multifactorial design, the authors cannot determine the potential interactions between all the factors investigated in their study, hence the analyzes of individual factors become inadequate. For example, attributing the difference observed in

pregnancy rates solely by the different measures of stimulation of respiration can be misleading. It is most likely that the differential response to the 2,4-DNP is due to the detrimental effects of the cryopreservation procedure on the sperm function, which is not limited to the respiration activity. Not to mention that the progressive motility, a factor that is generally accepted as being the most important factor affecting the non-return rates in AI, is similar between the semen samples with different stimulations of respiration by the 2,4-DNP.

ii.       In the written portion of the Results section, the authors state that the semen used in their study was in the acceptable range for sperm motility (lines 165-166), but they did not provide any comparison with what is reported in the literature or in the industry (cryopreservation in Russia maybe different than in other countries and lead to different percentages in sperm motility before and after thawing). I am not familiar with all of the species presented here, but I found the progressive motility for the fresh and cryopreserved bull semen low. For that reason, determining the viability along with the motility would have been desirable.

iii.     The authors state that there was a reduction of sperm respiration rates in all species after thawing, but no statistical data (p values) were presented in the Table 1, or in the text. Actually, all the statements made by the authors about the results presented in Table 1 are not supported by any statistical analyses (Lines 171-175). In addition, there are no results for the respiration rate of reindeer sperm (fresh and frozen) presented in the Table 1, and these missing data should be explained at the bottom of the Table.

iv.     In regard to the evaluation of the fertility, no controls in which the females were inseminated with non-treated frozen-thawed semen were performed. This control group is essential to determine whether or not the stimulation of respiration by the 2,4-DNP has an effect on the pregnancy rates observed, and if the differences observed are truly due to 2,4-DNP or something else, and to determine the effect of the females on the pregnancy rate. For instance, the authors did not mention what was the “normal” pregnancy rate for the sows used in the current study with cryopreserved semen of similar quality. Since the pregnancy rates presented in Table 2 are also very low, the same controls should have been performed with the cryopreserved bull semen.

v.       The authors state that the correlation coefficient is high for the fertility of boar sperm. I disagree; a correlation coefficient of 0.7 indicates variables that

are moderately correlated, not highly correlated (lines 181-182). Regarding the fertility experiments, I am wondering how they were practically carried out: did the stimulation of respiration by 2,4-DNP measure for each sample of boar semen (pool of ejaculates?), then only the samples having the three different measures displayed in the Table used for the inseminations? If so, how many females were inseminated with each semen sample? In other words, how many females contributed to the pregnancy rate for each pig group? What is the relevance of these pig groups anyway since the pregnancy rates are shown?

3)   The discussion is misleading and the authors made conclusions that are not supported by their actual data.

i.         The authors assumed that the semen from the reindeer mainly uses glycolysis to produce ATP solely because a low respiration response by the 2,4-DNP was observed (lines 234-236). What about other oxidative metabolism pathways such pentose phosphate pathway or the ones using fats as a substrate? Other methods using inhibitors specific to these different pathways should be considered, especially when not much is known, before concluding that the assessment of oxidative phosphorylation can serve as a biomarker for the quality of sperm and freezability potential.

ii.       The authors also attributed the damage to the electron transport chain by 2,4-DNP to the production of ROS, and inferred (mainly from previous studies) a lot of effects on the sperm physiology to the ROS. What is the relevance with their study since they did not measure the ROS production? Moreover, the motility among the 2,4-DNP treated samples was not affected. As the authors mentioned (lines 253- 254), predicting fertility based on sperm characteristics displayed by fresh and cryopreserved semen is very difficult because it involves so many factors. None of the data presented in the current study convinced me that the increase of respiration rate after adding the 2,4-DNP is (or could be) an appropriate test to assess the fertilizing capability of fresh and/or cryopreserved semen. In fact, the only conclusion that the authors could have reached (if they would have performed an adequate statistical analysis as I suggested) is that one or several steps during the cryopreservation procedure decreases the respiration rate, which probably lead to a decrease of stimulation of respiration by the 2,4-DNP.

There are other minor issues that should also be addressed by the authors, but since I rejected the current MS, I did not mention them in my review.

To conclude, this study assessed the respiratory activity of different domesticated species before and after cryopreservation, and examine the stimulation of respiration by the 2,4-DNP. Although the topic of the study was relevant, this manuscript is of poor quality in terms of scientific writing (grammar, format, consistency, etc.) and lack of scientific soundness, which hampers the reader to fully understand their methodology and results. Moreover, I strongly disagree with their main conclusions, which are based on data that were not adequately analyzed. Therefore, I will not recommend the MS vetsci-1794794 for publication in the special issue of Veterinary Sciences “Latest Advances in Basic Reproductive Research in Domestic Animals”.

Please let me know if you need any additional information.

Author Response

Thank you for the review

Reviewer 3 Report (New Reviewer)

This scientific manuscript gives a fair contribution to this scientific domain..  Some sentences are confuse and are  not easy to understand. Also in discussion some sentences are not relevant for this manuscript because the are general knowledge that we see in many papers. So, discussion must be shortened. English text is not of good quality. I think all this paper must be adapted to a short communication with the significant results obtained in this work. Some sentences that are in methods must be changes to discussion. Poor conclusions can be extracted from these work. Now i will go in some details. Why boar semen was not frozen in straws?. I an little confuse, because i see straws and pellets, see lines 103 and 129. Reading this work, we think that OXPHOS and the use of 2.4-DNP were only used  in thawed semen.. Only in discussion i detect  that it. Some sentences must be changed, from methods to discussion, namely  lines 134-138.. The details of semen preparation for CASA   and which semen parameters were studied  must be indicated.  See tables 1, what means in the third column Respiration rate nAO2/min?. What means?,. Confuse lines 212, anaerobic glycolysis (line 212) vs 210 Boar oxidative phosphorilation. After all?. Confuse text, lines 245-247; 188-190;  253-255; 275-277;  Why nine days, line 155?;  Lines 212-214,  249-252, not relevant.Also, table 3, is not important, because there are no significant results.  Finally a short communication is suggested.

Author Response

Thank you for the review

This manuscript is a resubmission of an earlier submission. The following is a list of the peer review reports and author responses from that submission.

Round 1

Reviewer 1 Report

In this paper, the authors carried out an experiment to assess the semen respiratory activity in different domestic animals and cocks before and after cryopreservation.

The aims and the design of the experiment are correct, but the explanation of the methodology lacks information (references).  The sections “Results” and “Discussion” can be significantly improved.

I am sorry, but I recommend the rejection of the paper.

General comments

Introduction

The introduction is correct and gives enough information to understand the aim of the experiments. However, some references should be added (see specific comments).

Material and methods

  • References about the protocols used to cryopreseve semen miss (see specific comments)
  • A Figure to explain better how 2.4-DNP work and how you measured it would be recommendable.
  • According to what you write, I understand the artificial insemination was only performed in cows, sows and hens, but the semen of reindeer and stallion was not used for this purpose. Could your clarify/specify it?

Results

  • The authors should provide the values acceptable for sperm morphology and motility and reference them. Moreover, why did you not measure the viability?
  • Also, the authors should give more information about what do exactly mean the values of stimulation of the respiration (i.e.: add a range of values with average results from previous referenced studies) to help the readers understanding easily the results.
  • Why do you not show the pregnancy results of sows in a Table as you do for cows and hens?

Discussion

  • The first paragraph is a contradictory. The authors say “Glycolisis predominates in bull and ram semen2” (Line 178) and then they write “Bull and rhesus monkey lie in between these extremes (Lines 182-183). Could you clarify it?
  • As I stated before, a Figure explaining how 2.4-DNP disrupts the respiratory chain would improve the paper.
  • References should be added (for example, in paragraph between lines 236-245).
  • In the last paragraph I do not understand the relation of sugars in seminal plasma, as you conducted and study using cryopreserved semen and there is no seminal plasma in it…

Specific comments

Introduction

  • Lines 33, 35, 45 and 50: please, add reference (s)

Material and methods

- Line 81: “Makler chamber”

- Line 84: please provide references about the protocols followed to cryopreserve the semen of the different species.

- Line 91: x10^6

- Line 94: 150 ml

Results

  • Table 1: Heading of the table!

Discussion

  • Line 179: “boar spermatozoa are”

Reviewer 2 Report

I enjoyed reading this manuscript! I have no doubt the readership will find it also interesting once revisions have been implemented. It should be suitable for publication following revision of the manuscript.

My key concerns are that the case number n is (understandably) relatively low for cattle and pigs, whereas they are adequate for poultry. Thus significance is weaker for mammals but high for birds. Can we really draw such general conclusions from these comparisons among species as the authors did?

Another key concern is that the authors argue that respiration is not correlated to fertility, but when boosted with 2,4-DNP, it is.  In their conclusion the authors are unclear and somewhat vague about the outcome and the interpretation whether respiration is required and or sufficient for fertilization rates. I would welcome a clearer/deeper discussion of gain and loss of function with the use of 2,4 DNP and what it means for the role of respiration in general.

It is unclear whether the polarographic method is novel in this mupblication or was prevoiusly described. In either case, it would be helpful to describe the method or its concepts in the introduction.

Title of the manuscript could be modified, for example: Assessment of Semen Respiratory Activity of Domesticated Species Before and After Cryopreservation: boars, bulls, stallions, reindeers, and roosters.

Other comments/suggestions/minor concerns:

Line 20 and line 148: replace power with strength?

Line 22 typo: respitation

Line 36 replace generate with require

Replace cock with rooster; (and cocks with roosters) throughout entire manuscript

Line 63: Please be more specific and list the scientific names with the specific breeds where applicable. Please list the source from where these animals were obtained, and or how they were maintained (husbandry conditions, including feeding).

Section 2.4. Semen Collection and Preparation should be subdivided in subsections for each species for better overview

Line 91 please superscript the 6 in 106

Line 94 please abbreviate mln (million) with 106

Line 111 and 152, typos: replace “.” with "comma" in 2,4-DNP. Fix throughout manuscript please! Also, in Table 2 column header.

Line 119 replace found with determined or calculated

Line 121 and 123 replace frozen with cryopreserved and thawed (or frozen and thawed)

Table 2 you could switch the columns for motility and respiration rate to maintain the same sequence as in Table 1. (Or change the order of columns in Table 1 to be more consistent with Table 2).

Table 2 legend: You could write P< 0.05 for a-f. Three times P< is redundant

Table column headers should be left aligned, readability is difficult otherwise

Line 162 typo: correlation

Lines 168 and 173 What method did you use to analyze/calculate correlation? What is the r and P value for respiration and fertilization versus respiration and 2,4-DNP stimulation (r=0.76, P<0.01)?

Lines 187 through 209 (approximately) might fit actually better to the Introduction, including the description of polarographic assay of oxygen concentration? Needs to be abbreviated a bit.

Line 261 add a line return before Our studies (new paragraph).

Line 266 I am confused by your last sentence, and you may need to reformulate your statement to state more clearly what you mean here. The test with DNP only shows that the respiration rate can be boosted with 2,4-DNP because the proton gradient of the inner mitochondrial membrane breaks down. However, you discuss that on its own normal respiration is not (well?) correlated with the fertilization rates in these species.  In spite of testing with DNP, somehow you leave the reader uncertain whether respiration is required or sufficient for fertilization. Could you please clarify this in a revision?